# Lack of harmonisation of greenhouse gases reporting standards and the methane emissions gap

Simone Cenci [1] ✉ & Enrico Biffis [2] ✉

Monitoring companies' contributions to climate dynamics and their exposure to transition risks requires accurate measurements of their non-carbon dioxide greenhouse gas emissions (non-$CO_2$ GHG). However, carbon accounting standards are not harmonised and allow for some discretion when converting emissions of different GHGs into $CO_2$ equivalent units, the currency in which carbon footprints are expressed. Focusing on methane, we build counterfactual harmonised standards using the latest IPCC Global Warming Potential (GWP) values over 100 years and estimate a cumulative gap in reported methane emissions of 170MtCO$_2$e ( ~6Tg) over a sample of 2864 companies. Changing the counterfactual from GWP$_{100}$ to GWP$_{20}$, as recently codified in certain jurisdictions and initiatives, increases the cumulative gap to 3300MtCO$_2$e ( ~40Tg). The gap only covers direct emissions and hence understates the extent of potential under-reporting across value chains. Overall, our study underscores the importance of global harmonisation of $CO_2$-equivalence standards to coherently track corporate GHG emissions and their exposure to transition risks.

Corporations play a crucial role in global decarbonisation efforts[1–3] but face substantial transition risks emerging from their business operations being exposed to changes in regulation, consumer preferences, technological innovation, and investors' pressure[4,5]. Their actions and transition plans are therefore systematically scrutinised by both policymakers and market participants, who have promoted greater transparency and accountability in reported GHG emissions and abatement plans[6–8].

To produce GHG emission disclosures, companies follow accounting frameworks that are largely based on the GHG Protocol[9], which has become the de-facto standard for carbon accounting, as it has inspired most reporting frameworks, such as those set by the Global Reporting Initiative (GRI) and the International Financial Reporting Standards Foundation (IFRS). Under the Protocol, companies disclose their aggregate GHG emissions in $CO_2$-equivalent ($CO_2$e) terms across different scopes, whereas disclosure of individual GHGs,

particularly those covered by the Kyoto Protocol, is recommended but not required.

In practice, the notion of $CO_2$-equivalence is pervasive, and market participants have grown accustomed to emission footprints and abatement targets reported in aggregate, $CO_2$e units[10]. Yet, different GHGs feature vastly different radiative efficiency and persistency profiles[11], with material implications for atmospheric oxidative capacity and ozone air quality. They therefore pose different challenges for climate risk mitigation[12], which have prompted calls for reporting and monitoring emissions in disaggregated and direct form[10,13,14]. Such changes would be of paramount importance to reliably monitor corporate contributions to global climate dynamics and national carbon budget calculations[15,16].

A particularly critical issue around disaggregated disclosure of individual GHGs is a lack of harmonisation of practices and guidelines, which allow for optionalities in reporting choices. For example, for

[1]Institute for Sustainable Resources, University College London, London, UK. [2]Centre for Climate Finance and Investment and Department of Finance, Imperial College Business School, London, UK. ✉e-mail: simone.cenci@ucl.ac.uk; e.biffis@imperial.ac.uk

companies disclosing individual, disaggregated, GHGs, the GHG protocol recommends reporting emissions in $CO_2$e terms based on the GWP computed over a 100-year horizon ($GWP_{100}$) sourced from the latest IPCC Assessment Report's (AR), but acknowledges the fact that corporates "may choose to use other IPCC Assessment Reports."[9]. Most corporations worldwide are not legally bound to follow the GHG protocol guidelines, and other choices and recommendations exist. Most countries and environmental agencies around the globe, such as, for example, the United States Environmental Protection Agency (EPA), as well as well-established reporting standards, such as the GRI, recommend the use of $GWP_{100}$, but not necessarily from the latest IPCC AR[17,18]. Indeed, some jurisdictions even recommend the use of older ARs[19], whereas the IFRS S2 Climate-related Disclosures requires the use of the latest $GWP_{100}$[20]. On the other hand, some jurisdictions (e.g., the US States of New York and Maryland) and initiatives (e.g., the Global Methane Pledge) have opted for the use of GWP based on a 20-year horizon ($GWP_{20}$) to pursue aggressive emission reduction plans, by putting greater emphasis on the abatement of short-lived gases such as methane[21,22].

The choice of appropriate metrics allowing comparability of different GHG emissions has received considerable attention in the scientific community. The IPCC suggests that $GWP_{100}$ strikes a reasonable balance between the warming effects of climate forcers and their different lifetimes, but also acknowledges that the choice has no unique grounding[23]. Some authors[24] suggest that the time horizon can be tailored to pursue alignment with specific temperature goals (e.g., a 1.5°C warming and thereby a 24-year time horizon), whereas other authors link the time horizon to the discounting of climate damages with rates consistent with various economic studies[25]. Other authors[21] suggest that a 20-year time horizon should be used to pursue aggressive abatement of short-lived climate forcers to slow the rate of warming over the next few decades and "buy time". These views stand in contrast with those who emphasise the need to focus on long-lived climate forcers, as the warming they bring about is, for all practical purposes, irreversible[11].

In this article, we examine how companies navigate fragmented guidelines and recommendations on non-$CO_2$ reporting by analysing a large-scale dataset of corporate emissions across multiple countries and sectors. We study the implications of lack of harmonisation by focusing on methane, which is the second most important contributor to historical warming after $CO_2$, with an effect that is comparable in scale (0.5° [0.3°-0.8°] versus 0.8° [0.5°-1.2°] for $CO_2$[26,27]) and is a critical driver of short-term warming dynamics[28,29], making it instrumental for meeting the 1.5°C target[30,31]. Moreover, methane's GWP varies substantially with the time horizon considered, due to its short lifetime[11,24]. Indeed, the GWP indicated over time by various assessment reports increased from 21x to 28x for the 100-year horizon and from 56x to 84x for the 20-year horizon (see table ST2 in the Supplementary Information). In other words, each ton of emitted methane could have contributed to a carbon footprint of between 21 and 84 $CO_2$e tons, depending on the assessment report and time horizon considered, a wedge ranging between 33% and 300%.

We quantify the implications of lack of harmonisation by comparing reported methane emissions with hypothetical, counterfactual emissions computed based on systematic applications of the latest available IPCC value for $GWP_{100}$ and $GWP_{20}$. The approach allows us to compute a methane emission reporting gap —the gap between emissions reported and those expected under the harmonised standard— and to investigate how companies in different sectors tend to align or deviate from the counterfactuals. We then quantify the economic relevance of these results by using carbon prices. Overall, our results provide strong support for initiatives pursuing global harmonisation of non-$CO_2$ accounting, as well as direct reporting of differentiated GHG emissions in native unites of measure.

## Results

### Heterogeneity in emission metric selection

To document the behaviour of companies faced with fragmented reporting guidelines, we use data from CDP (formerly, the Carbon Disclosure Project) questionnaire between 2014 and 2023. CDP is a leading organisation for voluntary climate disclosures and a key benchmark for corporate emission data. Here, we collect data from self-reported Scope 1 emissions, by taking into account the breakdown between $CO_2$ and non-$CO_2$ gases alongside the GWP values used to report them in $CO_2$e units (see "Methods" for further details on the dataset). We focus on Scope 1 emissions because companies rarely report Scope 2 and Scope 3 emissions differentiated by individual GHGs. Moreover, Scope 3 emissions entail additional degrees of optionality as they can be estimated using different methodologies[9,32]. Finally, focusing on Scope 1 emissions reduces the risk of double counting.

Based on data made available to us by CDP, our full sample covers 14077 companies from 120 countries and across all economic sectors (see Figure S1 for a summary statistics of the dataset). We find that the number of companies responding to the climate change questionnaire has dramatically increased over time (from 1825 in 2014 to 10866 in 2023), but the number of those that have explicitly reported their methane emissions and made explicit the metric sourced from an IPCC AR is substantially lower (from 371 to 2072, Fig. 1a).

Overall, this subsample, which will be our main focus, includes 2864 companies responsible for approximately 65% of the total Scope 1 emissions within the CDP universe, which includes ~8500 companies reporting non-zero Scope 1 values. Based on their reported information, the sample covers 11% and 2.5% of global GHG and methane emission (as available from the EDGAR inventory), respectively. In terms of economic relevance, the sample covers at least 40% of global revenue and market capitalisation. Most of the companies in our sample are from North America, China, Europe, Brazil and Japan. However, we also cover other geographies as shown in Figure S2 in the Supplementary Information.

The GWP used on average by this sub-sample of companies (dotted line in Fig. 1a) is systematically lower than what would be expected under the most recent IPCC guidance available in any given reporting year along the sampling period (AR5 from 2014 to 2021 and AR6 from 2022 to 2023). The differentiation in the most recent GWP values appearing in 2022 and 2023 is due to the fact that AR6 provides separate methane GWP values for fossil and non-fossil sources. Deviations of the average emission metric from the latest available values are mostly driven by heterogeneity in the choice of the reference AR used to source the relevant GWP (Fig. 1b). Interestingly, there are companies that for their 2023 reports still use GWP values published in the Second AR of 1995. On the other hand, most companies in our sample (82%–90%) rely on a 100-year time horizon for the GWP, consistently with the main recommendation of the GHG protocol (Fig. 1c).

### Accounting optionality and its impact on methane emission monitoring

The deviation of the average GWP of the sample from the most recent values available in the reporting year (Fig. 1a) suggests that reported emissions are lower than they would be on aggregate under a hypothetical harmonised standard (e.g., the most recent $GWP_{100}$). The actual magnitude of the gap between reported and counterfactual emissions, however, depends on the methane specifically emitted by deviating companies. Here, we measure a methane conversion gap as the difference between methane emissions reported by each company and those that would be reported under a counterfactual, harmonised emission metric, which we denote by $GWP_{IPCC, H}$. The pair (IPCC,H) identifies the counterfactual chosen by including the benchmark IPCC report and time horizon H for the GWP metric (see The methane

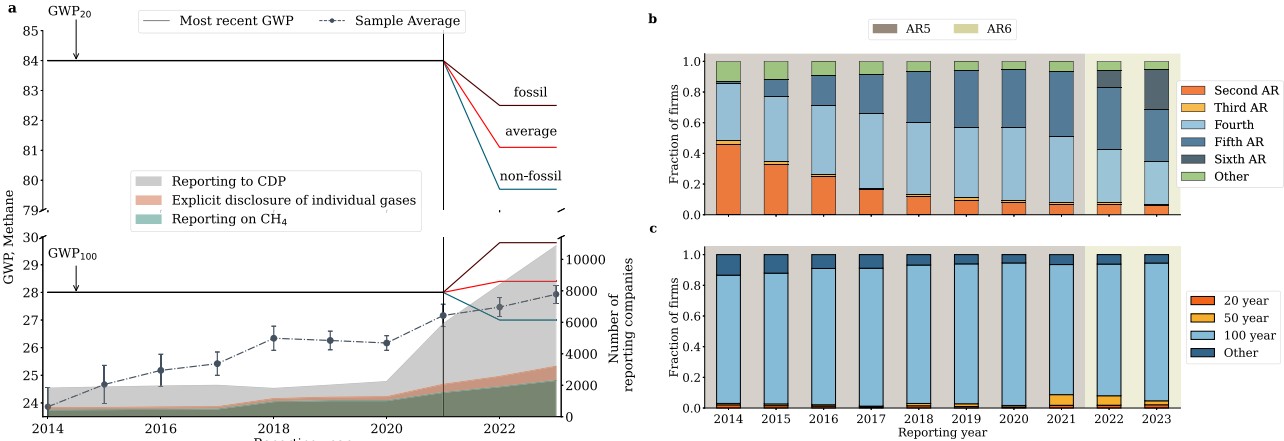

**Fig. 1 | Emission metrics. a** shows the yearly number of companies in our sample (grey, bottom right *y*-axis), the number of companies that report differentiated Scope 1 emissions by individual GHGs (orange, bottom right y-axis) and the number of companies reporting methane emissions (green, bottom right y-axis). The dash-dotted line (left y-axis) shows the average GWP values used by companies to convert methane emissions into $CO_2$e units; the error bars show the standard errors, N = 9375. The solid lines show the $GWP_{100}$ and $GWP_{20}$ from the latest Assessment Report (AR), including the fossil and non-fossil GWP values from AR6. The solid vertical lines denote the transition year from AR5 to AR6. **b** and **c** show yearly distribution of GWPs in the sample by Assessment Report (top) and GWP time horizon (bottom). The proportions are computed based on the 3248 companies reporting methane emissions (N = 10578).

conversion gap). The conversion gap is then given by

$$\Delta_{CH_4}^{H}(t) = \mathcal{E}_{CH_4}^{reported}(t)\left(1 - \frac{GWP_{IPCC, H}(t)}{GWP_{reported}(t)}\right), \quad (1)$$

where we note that the counterfactual is time-varying, as it tracks the release of new guidance available in each reporting year *t*. A negative gap entails that emissions are underreported relative to the counterfactual. In the following, we use the most recent $GWP_{100}$ as a baseline counterfactual since this is consistent with GHG protocol's guidelines and the IPCC's default reporting choice aimed at balancing the contribution of short-lived and long-lived gases. We have $GWP_{100}=28$ and $GWP_{100}=28.4$ during the time intervals [2014, 2021] and [2022, 2023], respectively. However, we also consider the most recent $GWP_{20}$, as certain jurisdictions and initiatives have recently opted for such a metric and some authors believe it to more effectively enable alignment with the targets of the Paris Agreement[24]. We have $GWP_{20}=84$ and $GWP_{20}=81.1$ during the time intervals [2014, 2021] and [2022, 2023], respectively.

The conversion gap indicates that lack of harmonisation in reporting standards has resulted in a cumulative underestimation of methane emissions within our sample amounting to 170MtCO$_2$e for H=100. When considering the case of H =20, the extent of the cumulative underestimation rises to 3300MtCO$_2$e (Fig. 2a, e see "Methods"). In relative terms, methane emissions would have been approximately 1.12 higher than reported under the $GWP_{100}$ counterfactual, and 3.3 times higher if reporting had been instead harmonised to the 20-year time horizon (figure 2 inset **c** and **g**).

Notably, insets **b** and **f** in Fig. 2 show that the impact of divergence from counterfactuals is not uniform within and across sectors, with companies in the Energy, Utilities, Material, and Consumer Staples sectors largely dominating the conversion gap. Importantly, while the average GWP of the sample converges towards the latest IPCC values throughout the observation period (Fig. 1a, sector-specific heterogeneity persists over time. In particular, sectors that are instrumental in enabling the low-carbon transition, such as Energy and Utilities, feature a substantial gap even during the last few years of observation (Fig. 2d, h). For companies in those sectors, methane is a sizeable contributor to total Scope 1 emissions (up to an average of 10%, Fig. S5 in the Supplementary Information). Hence, loose reporting guidance is

particularly problematic since the choice of the GWP can distort aggregate GHG footprints, making them unreliable for assessing the effectiveness of climate policy and misleading for measuring the climate footprint of investors' portfolios.

Focusing again on $GWP_{100}$ as our main baseline, we find systematic differences between companies that follow the guidelines and those that do not. Companies diverging from the guidelines in either the time horizon or the reference AR, are, on average, larger, more mature and hold a greater proportion of tangible assets in their books (i.e., property plants and equipment). Importantly, the probability of deviation from accounting standards is higher for companies that implement a lower number of carbon management activities as reported to the CDP climate change questionnaire, whereas it decreases with the time elapsed since the latest $GWP_{100}$ release (see table ST1 and section B in the Supplementary Information).

## Implications of longitudinal inconsistencies

Our analysis reveals important cross-sectional variability in the GWPs used by companies in our sample, but does not inform us as to whether companies are consistent with their own choices on a year-on-year (YoY) basis. To investigate the longitudinal consistency of each company's disclosure, we analyse the methane emission reporting choice over time.

Excluding companies switching towards the latest available counterfactual GWP, approximately 3% of the companies in our sample with at least two consecutive years of observations feature at least one switch from $GWP_{100}$ to $GWP_{20}$ or viceversa (Fig. 3a, d). This subsample represents 0.5% of the companies reporting non-zero Scope 1 emissions, but contributes to ~ 1% of those emissions. Switching from $GWP_{20}$ to $GWP_{100}$ leads to YoY rate of change in emissions that is lower than what it would have been under a constant 100-year or 20-year counterfactual (see Intertemporal choices), making the change in metric equivalent to emission abatement from a reporting perspective. In our sample, this change leads to an average emission abatement of 200% relative to the counterfactual (red bars in Fig. 3b, e). Switching from $GWP_{100}$ to $GWP_{20}$ clearly has the opposite effect, but the overall impact on reported emissions is substantially smaller. Hence, time-variation in the choice of emission metrics leads to overall underestimation of changes in emissions reported by companies in our sample.

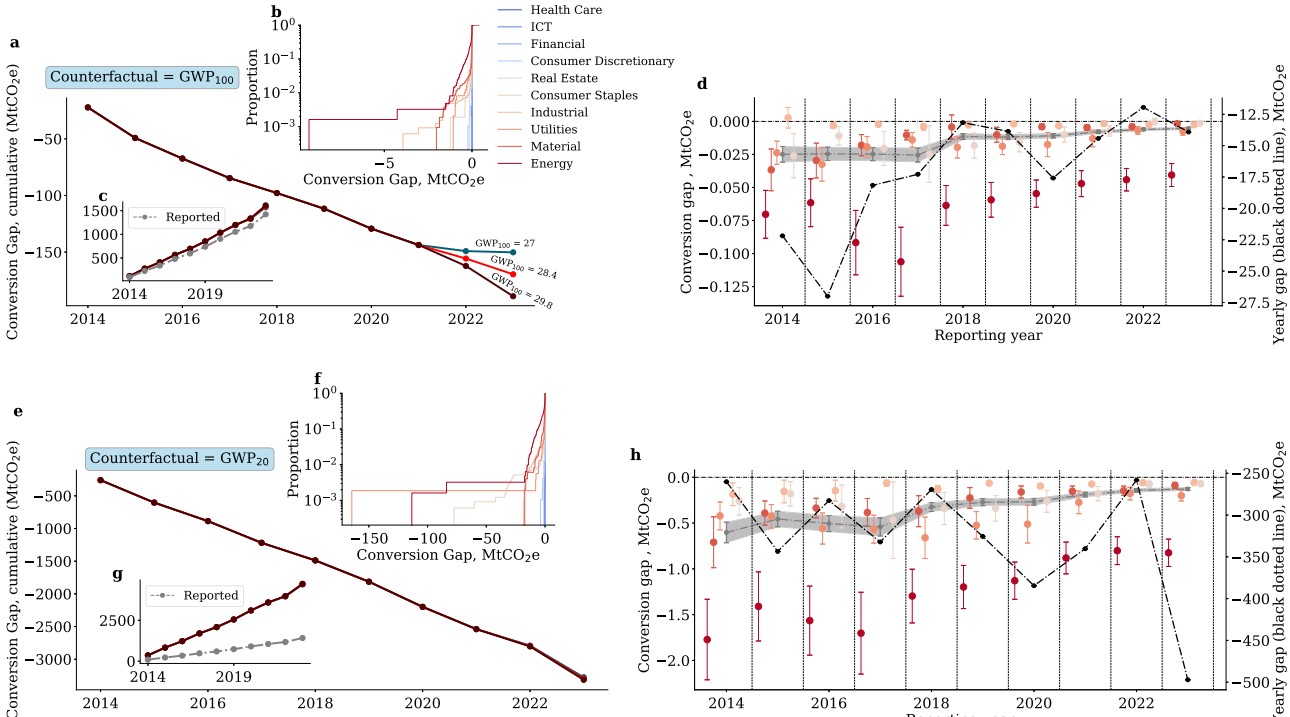

**Fig. 2 | The methane conversion gap. a, e** shows the cumulative value of the methane conversion gap over time, i.e., the sum of the yearly differences between reported methane emissions and emissions computed under the most recent $GWP_{100}(GWP_{20})$. The trifurcation of the solid lines from year 2021 shows the gap under the fossil and non-fossil GWPs from AR6, as well as their average. **b, f** shows the cumulative distribution of the methane conversion gap across companies under the most recent $GWP_{100}(GWP_{20})$. **c, g** shows the cumulative value of methane emissions as reported and as expected under the most recent $GWP_{100}(GWP_{20})$ (N =

9375).**d, h** left y-axis shows the average methane conversion gap across key sectors under the 100-year (20-year) counterfactual. The error bars are standard errors of the mean (N = 8713); low methane emission sectors and the bottom 1% of the gap distribution are excluded from the visualisation and the estimation of the averages. The grey dot-dashed line shows the sample averages across time, whereas the grey shading shows the standard errors. The black dotted line in the right y-axis shows the total gap over the sample, including all sectors, by year.

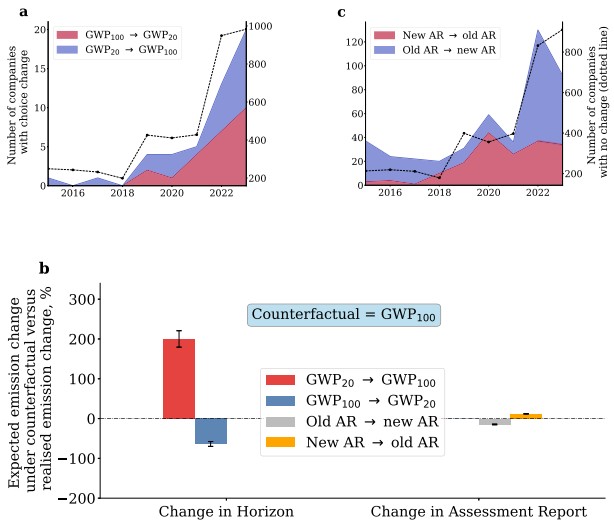

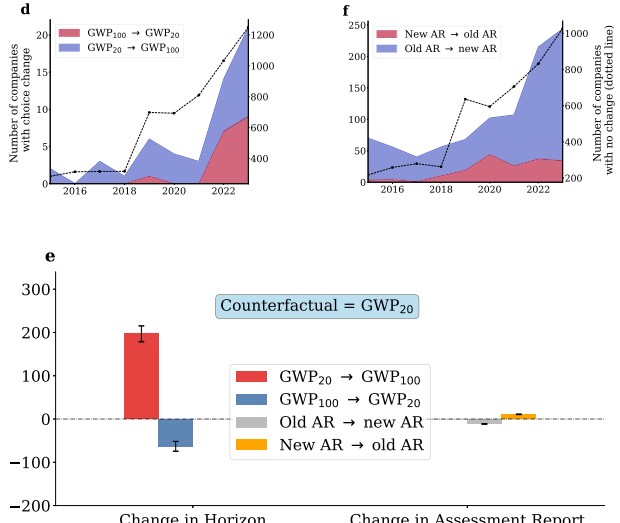

**Fig. 3 | Intertemporal choice. a, d** and (**c, f**) show the number of companies adopting a different emission metric (left y-axis) compared to the previous year's choice of Assessment Report, excluding companies adopting the counterfactual (the most recent $GWP_{100}$ and $GWP_{20}$ in **a, c** and **d, f** respectively). The dark dotted lines show the number of companies that do not change metrics (right y-axes). Panel **b** (**e**) shows the implication of companies' choices on year-on-year changes in emissions under the $GWP_{100}(GWP_{20})$ counterfactual, as in (3). From the leftmost

red bar in (**b**) to the rightmost yellow bar in (**e**), the sample includes N=48,48,504,344,74,34,1316,344 observations which only represent companies that have changed metrics over consecutive years and exclude those that have switched towards the counterfactual GWP. Positive values imply realised rates of change in emissions that are lower than what they would have been had the company consistently adopted the counterfactual GWP. Error bars represent standard errors of the median.

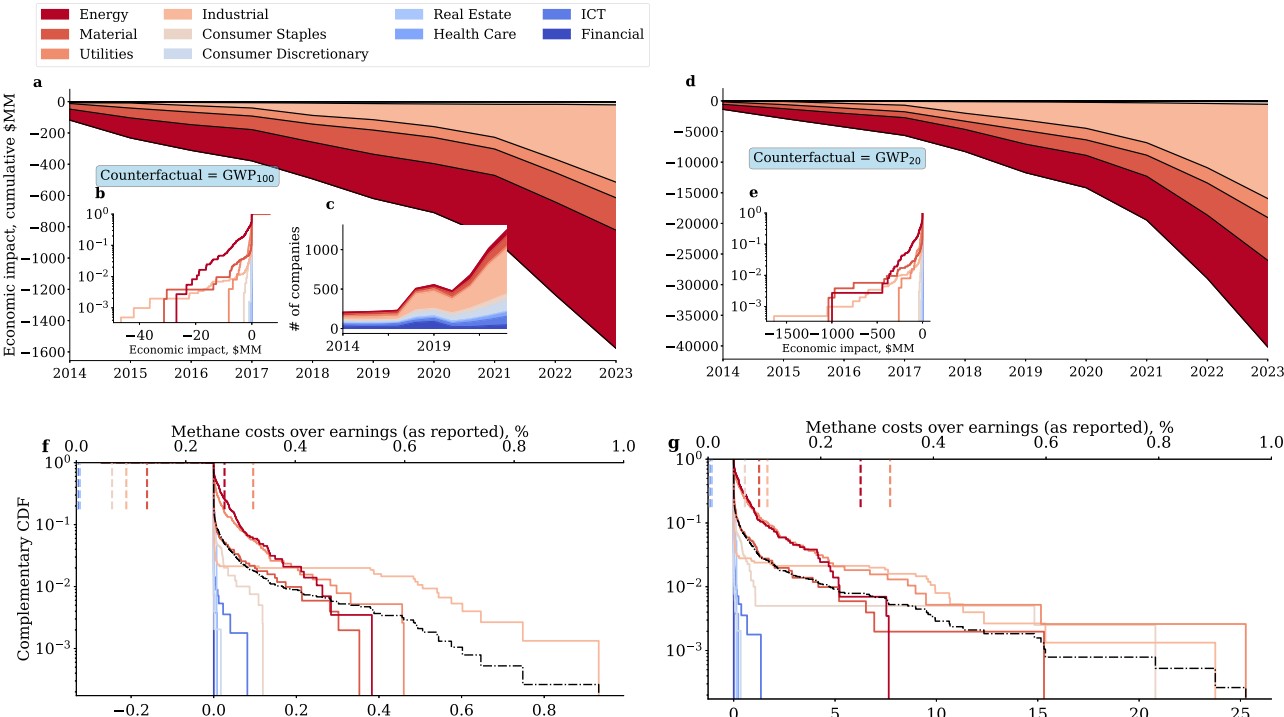

**Fig. 4 | Economic implications of the methane conversion gap. a**, **d** show the cumulative sum of the yearly total difference between cost of methane under reported and counterfactual emissions by sector (N=5380). **b**, **e** show the cumulative distribution function of the economic impact across companies and years under the $GWP_{100}$ and $GWP_{20}$ counterfactuals, respectively. **c** shows the number of companies in the sample by year and sector (the number of companies is the same

in (**a** and **d**). **f** and **g** show the complementary cumulative distribution function (1-CDF) of earnings at risk under the $GWP_{100}$ and $GWP_{20}$ counterfactuals, respectively (N=4265). We express EAR in percentage points and therefore multiply (5) by 100. The black line show the complementary cumulative distribution function across all sectors. The dotted vertical lines at the top of the figures denote the average cost of methane (as reported) as a fraction of EBITDA.

A substantially larger proportion of companies not switching towards the counterfactual and with at least two years of consecutive observations (26% under the $GWP_{100}$ counterfactual and 42% under the $GWP_{20}$, representing 15-30% of CDP Scope 1 emissions) changed the reference AR at least once during the observation period. Most changes are associated with updates towards more recent ARs (Fig. 3c, f). However, we also observe switches in the opposite direction (from most recent, to older ARs), and more of these in recent years as opposed to the beginning of the sampling period. As for the impact on reported versus counterfactual emissions, updating towards a more recent AR implies an increase in YoY changes in reported emissions (relative to the counterfactual). In our sample, this difference is statistically significant but small (grey bars in Fig. 3b, e). The opposite is true for backward changes in ARs, but the net effect is indistinguishable from zero. Looking at the sectoral and geographical composition of the sample of companies that changed their GWP choice over time, we have not found any systematic bias.

**Economic impact of optionality**

Harmonisation of methane accounting standards is required to provide investors with accurate and comparable measurements of GHG emissions as well as aggregate emissions. As the methane conversion gap we estimate is negative (methane emissions are under-reported), the economic implications of a transition to harmonised standards cannot be overstated. As a proxy for this transition risk channel, we quantify the earnings at risk associated with the methane emission gap. We collect global carbon price data from active ETSs from the World Bank and estimate the economic impact of reporting optionality by taking the product of the yearly methane conversion gap and the price of carbon as of the year of reporting (see Economic impact and earnings at risk).

We were able to collect carbon price data for 1718 companies, as not every country in our sample has an active ETS during the observation period. The sub-sample is still relevant, as it covers ~65% of total Scope 1 emissions of the 2864 companies analysed in this study. However, several large methane emitters, particularly from the Consumer Staples sector (which includes agriculture activities) are not included. We find that the methane conversion gap of the sub-sample is associated with a cumulative economic loss of approximately $1.6bn ($40bn) under the $GWP_{100}$ ($GWP_{20}$) counterfactual over the observation period (Fig. 4a, d). Notably, the economic loss in our sample is strongly driven by companies in the Energy, Utilities, and Material sectors where company-level yearly losses can amount to as much as $1bn (Fig. 4b, e). Looking at the geographical heterogeneity of the economic impact, we find that North American, Western and Southern European companies are most exposed to economic losses (Figure S9).

We measure transition risk as the portion of earnings "consumed" by the carbon cost of methane emitted. We then take the difference between such a measure computed under the counterfactual and for reported emissions (see Economic impact and earnings at risk). The higher the difference, the greater the fraction of earnings that would be at risk should methane emission reporting follow harmonised standards.

Reported methane costs represent in general a small portion of earnings—approximately 0.2–0.4% in high emitting sectors and less than 0.1% in any other sector (dotted lines in the top x-axis in Fig. 4f, g). However, we observe substantial heterogeneity in the distribution of transition risk across sectors, with Energy and energy-intensive sectors facing considerably greater earnings at risk than any other sector. Indeed, we find that in high-emission sectors

companies face transition risks that can be 1.4 times larger than currently reported under an harmonised $GWP_{100}$, and as much as four times larger than currently reported under a change of norm to a harmonised $GWP_{20}$counterfactual. Looking at geographical heterogeneity, we find that North American, Western and Northern European companies have a greater exposure to transition risks (Figure S10).

## Discussion

Monitoring corporate contributions to global climate dynamics and resolving ambiguities in temperature target alignment require accurate measurements of non-$CO_2$ emissions from business operations. Loose guidelines and regulations around reporting requirements allow companies to choose whether and how to disclose their non-$CO_2$ emissions. This makes, in turn, corporate emission footprints and abatement plans challenging to monitor and compare. Using data from the CDP climate change questionnaire, the most comprehensive source of non-$CO_2$ emission reporting to date, and focusing on methane, we have documented the extent to which companies deviate from well-established baselines as well as the economic implications of reporting optionality under $GWP_{100}$ and $GWP_{20}$ counterfactual harmonised standards.

Our results show that only a small portion of companies in our sample ( ~20%) actively disclose methane emissions using the latest $GWP_{100}$(Figs. 1 and Figure S4). While most companies(82–90%) still use a 100-year horizon, sourcing it from older ARs induces deviations from the harmonised standard that have a substantial impact on the sample's total emissions (Fig. 2). Indeed, under a harmonised $GWP_{100}$ counterfactual, cumulative Scope 1 methane emissions in the sampling period would have been $170MtCO_2e$ higher than actually reported. This represents approximately 12% of total reported Scope 1 methane emissions. The total amount of methane emissions that are not accounted for in the CDP reporting programme under the $GWP_{20}$ counterfactual is substantially larger, i.e., ~$3300MtCO_2e$ (approximately 230% of total reported Scope 1 methane emissions). While the $GWP_{20}$ metric is not mainstream in emission reporting guidelines, it is gaining traction in jurisdictions pursuing aggressive emission reduction plans by targeting short-lived gases[21,24]. Hence, it is important to understand the impact of under-reporting of historical corporate emissions relative to those initiatives.

To contextualise our findings, we note that the cumulative methane conversion gap we have estimated under our two counterfactuals ranges between 1.3x ($GWP_{100}$) and 10x ($GWP_{20}$) of the 2022 methane emissions of countries such as Australia, Canada, and Saudi Arabia. We recall that these values only refer to the Scope 1 methane emissions of a relatively small, albeit economically significant, sample. This suggests that lack of harmonisation around methane emission reporting might result in substantial underestimation of the private sector's climate footprint after extrapolation to the wider universe of companies and their indirect emissions.

The impact of reporting optionality on the monitoring of corporate non-$CO_2$ emissions is highly heterogeneous in our sample. Indeed, most of the methane conversion gap has historically been, and still is, driven by companies in the Energy sector, where mean (median) methane emissions add up to ~10% ( ~5%) of total emissions (Figure S5). Our findings therefore highlight the need for robustifying their emission estimates in order to effectively support emission reduction targets, which have already been documented to be misaligned with the targets of the Paris Agreement[8]. As we limit ourselves to analysing Scope 1 emissions, we cannot assess the full impact of the methane reporting gap on the total footprint of those companies, but extrapolation across all scopes can be expected to make the overall impact considerably more sizeable. Future research should address this dimension by including more granular data on Scope 3 emissions broken down into individual GHGs. This is a challenge at the moment, given the further latitude allowed by accounting standards in estimating indirect emissions[32].

Moving beyond cross sectional statistics, our study reveals interesting dynamics in GWP choices. Notably, between 40 and 700 companies (depending on the choice of $GWP_{100}$or $GWP_{20}$as a counterfactual) change their emission metrics over time by opting for a shorter time horizon, even if for a small period of time. This may result in inflation of their emission, even if only temporarily. We cannot observe or infer the rationale for this choice, but several drivers might be at play. For example, companies may switch metrics in anticipation of changes in regulation or they may wish to inflate the offsetting potential of abating certain short-lived gases, as emission permits are traded and retired on a $CO_2$ equivalent basis. Companies may also simply rely on emission estimates outsourced to providers that have discretion in how to interpret reporting guidelines. The lack of sectoral and geographical bias in corporate choices suggests that these dynamics are mostly driven by internal factors as opposed to external influences. Hence, harmonisation of emission reporting standards could represent an important disciplining device to ensure comparability of reported emissions and mitigate greenwashing concerns.

In this study, we have not addressed the question of why companies might deviate from guidelines or choose a particular emission metric over other ones. We do not make any causality claims, neither in the cross-section nor in the temporal dimension. As discussed throughout the manuscript, there could be multiple reasons behind those choices, from regulatory requirements to country-level policy recommendations or strategic reporting behaviour, as well as simple lack of engagement. We also do not quantify the uncertainty around the estimates because companies do not share the raw data supporting their emission estimates. The uncertainty surrounding certain estimates is sometimes disclosed in sustainability reports but is typically limited to aggregate GHG emissions and provided in unstandardised format. This makes gauging the uncertainty of estimates at individual GHG level extremely challenging.

Optionality in the choice of emission metrics has important implications for investors assessing their portfolios' exposure to transition risks, as proxied, for example, by carbon price risk. Indeed, we have found that if companies were to follow harmonised standards, the cost of offsetting their methane emissions would represent a sizeable portion of their earnings. The effect of a change in reporting norm is particularly evident in the Energy and energy-intensive sectors, in which, under the $GWP_{20}$ counterfactual, earnings at risk can be as high as four times those implied by reported methane emissions. These estimates are based on ETS carbon prices applied also to companies that are currently not subject to them. We note that companies participating to ETS are subject to tighter regulations in their emission computation, i.e., emission reporting regulations are harmonised within each ETS. Hence, given the fast growth of compliance markets worldwide, the economic costs of transition risks associated with the implementation of harmonised standards cannot be overstated.

Overall, our results contribute to the emerging literature on carbon accounting standards for non-$CO_2$ gases. As pointed out by some authors[33], recent efforts have aimed at improving the coherence and consistency of accounting within and across different scopes[32,34,35], but have taken the unit of account for granted. In this work, we have focused on methane to demonstrate the need for consistent reporting of non-$CO_2$ emissions broken down by individual GHGs measured in native units of emission. Reliable monitoring of corporate emissions and associated abatement plans can only be implemented by moving beyond reporting standards relying on loose or heterogeneous notions of $CO_2$-equivalence. Indeed, without global harmonisation of GWP standards, $CO_2$-equivalence may be misleading when comparing different corporate emissions and may be vulnerable to manipulation, due to the heterogeneity in emission abatement costs faced by different sectors. More prosaically, use of an inconsistent unit of measure

seriously undermines accounting exercises that are so crucial in structuring and monitoring alignment with climate goals. The global harmonisation of direct reporting of disaggregated GHG emissions would also allow us to account for the different impacts of short-lived and long-lived climate forcers on global temperature[13], therefore enabling the accurate tracking of corporate contributions to global climate dynamics, particularly in sectors that are instrumental for the transition to a low-carbon economy.

We recommend that institutions and initiatives such as CDP explicitly require companies to disclose non-$CO_2$ emissions in differentiated format and report them either in native units of measure or under a harmonised GWP (e.g., the most recent $GWP_{100}$ released by the IPCC) or following a double reporting standard[36,37] (i.e., reporting using both $GWP_{100}$ and $GWP_{20}$). We also recommend that all organisations that track corporate emissions and their alignment with global climate targets, such as the Science Based Target initiative, require companies to disclose individual GHG emissions and set targets either using a full mix of GHG under harmonised standards or by individual group of climate forcers, as for example recently proposed by some authors[10]. This is to ensure a level-playing field and avoid that companies take advantage of reporting optionality to boost their climate performance. This is also essential to assess the contribution of the corporate sector GHG mix to global temperature rises[13,14,29].

Finally, as mandatory reporting requirements emerge throughout the globe through, for example, the implementation of new ETSs and the addition of new GHGs to existing ETSs, it is vital that new regulations are developed and implemented by ensuring harmonisation of reporting standards across jurisdictions and geographies.

## Methods

### Data
We collected data from the CDP Climate Change questionnaires from 2014 to 2023 as available from our academic license. From each questionnaire, we extract companies' identifiers (company name, account number, ISIN and Ticker symbols) and metadata, i.e., country, sector, industry, and answers' submission date. Then, we filter out observations that refer to emission data from previous fiscal years (e.g., from the 2015 questionnaire, we only retain companies that report data on the fiscal year ending in either 2014 or 2015). Finally, for those companies reporting the breakdown of Scope 1 emissions across individual GHGs, we extract the gas name, emission value (in $tCO_2e$), and the GWP choice.

We focus on Scope 1 because Scope 2 and 3 emissions are rarely disaggregated by individual GHGs. However, even for Scope 1 emissions, not every CDP respondent discloses emissions breakdowns and only a subset of those that disclose them explicitly report methane emissions and the GWP value used for the conversion. We group all companies that report methane emissions but do not explicitly disclose an IPCC emission metric in a separate category denoted as "Other"; see, for example, Fig. 1b, c. Some of these companies refer to country guidelines (that might still be based on the most recent GWPs), others use internal factors, and others do not report this information in English (see table ST3 for examples of GWP choices within the "Other" category).

It is important to note that several companies in the "Other" category might still have disclosed methane emissions using an IPCC value, even if they have not explicitly chosen that value from those provided by CDP in a drop-down list. As these values cannot be validated without considerable risk of misclassification, we remove them from the main analysis. The only exceptions are companies that select the "Other, please specify" option and then indicate an IPCC value following the same structure adopted by the CDP default options (e.g., "IPCC Fifth Assessment Report (AR5 - 100 year)"). In our analyses we match the text after the "Other, please specify" string and include these companies. We also drop companies disclosing emissions using a 50-

year time horizon, which is not a standard metric from the IPCC but an option provided by the CDP questionnaire. As shown in figure S1, these filters have a marginal impact on sample sizes across the years. Finally, some companies report methane emissions together with the emissions of other gases (e.g., "Other, please specify: CH4, N2O and HFC", "Other, please specify: CH4 (methane), N2O (nitrous oxide)"). We do not include these companies in our sample.

In terms of industry classification, we note that the raw data report it following inconsistent standards across the years. We therefore manually map industries and primary activities into the Global Industry Classification Standard (table ST4). Similarly, we map countries into geographical regions following the United Nations' SDG framework. Following the S&P geography classification scheme, we further aggregate data into three macro-regions (Europe, North America, Asia-Pacific) with one additional global region representing the rest of the world (table ST5). Aggregation is essential to guarantee enough yearly data for the sample statistics. To calculate the economic relevance of our sample we used data for approximately 53000 companies from Compustat. Global market capitalisation data are from the World Bank. Exact values of market share cannot be calculated because some of the companies in the sample are privately owned and other were lost in the matching process. Overall, the economic statistics are calculated over a sample of 1751 companies.

### The methane conversion gap
For every company in the fully disclosing subset we calculate the counterfactual emission values under the most recent $GWP_{100}$ and $GWP_{20}$ from the IPCC report available at the time of disclosure, i.e., AR5 from 2014-2021 and AR6 in 2022 and 2023. The counterfactual emissions for company $c$ in year $t$ are calculated as $\mathcal{E}_{CH_4, c}^{H, counter}(t) = \mathcal{E}_{CH_4, c}^{reported}(t) \frac{GWP_{IPCC, H}(t)}{GWP_c^{reported}(t)}$, where H is either 20 or 100. The associated methane conversion gap is then given by $\Delta_{CH_4, c}^{H}(t) = \mathcal{E}_{CH_4, c}^{reported}(t) - \mathcal{E}_{CH_4, c}^{H, counter}(t)$. The GWP values used in the calculations are reported in table ST2 in the Supplementary Information. The cumulative conversion gap shown in Fig. 2a is simply the cumulative sum along previous years and across companies' yearly conversion gaps. In other words, for each year $T$ between 2014 and 2023, we have:

$$\Delta_{CH_4, cum}^{H}(T) = \sum_{t \le T} \sum_c \Delta_{CH_4, c}^{H}(t) \qquad (2)$$

### Intertemporal choices
The implications of intertemporal choices are estimated as follows. For every company that changes emission metric from the previous year and does not re-aligns itself with the counterfactual we calculate two quantities: (1) the percentage change in reported emissions between year $t-1$ and year $t$ ($\mathcal{E}_{CH_4}^{reported}(t)/\mathcal{E}_{CH_4}^{reported}(t-1) - 1$) and (2) the percentage change in emission that would have occurred had the company followed the counterfactual in both $t-1$ and $t$ ($\mathcal{E}_{CH_4}^{counterfactual}(t)/\mathcal{E}_{CH_4}^{counterfactual}(t-1) - 1$). The first term is a function of both emission reduction capabilities and change in emission calculation methodologies. The second term is simply a function of emission reduction capabilities except for changes occurring upon publication of AR6, when the GWP value of the counterfactual changes as well. We then take the difference between the two and the median over all companies, c, and time periods, t, ($\langle \cdot \rangle_{t,c}$, we use the median because the statistics is calculated over a relative small sample with a skewed distribution). That is:

$$r = 100 \times \left\langle \frac{\mathcal{E}_{c, CH_4}^{counterfactual}(t)}{\mathcal{E}_{c, CH_4}^{counterfactual}(t-1)} - \frac{\mathcal{E}_{c, CH_4}^{reported}(t)}{\mathcal{E}_{c, CH_4}^{reported}(t-1)} \right\rangle_{t, c}, \qquad (3)$$

where $t$ and $c$ range over the set of all time periods and of all companies whose emission metric or AR source deviate from the previous year choice, respectively. The set excludes observations that switch to the most recent counterfactual. The statistics $r$ is calculated independently for any change of time horizon and assessment report and it is shown in the bar plots in Fig. 3b, e. Positive values of $r$ imply rates of change in emissions that are lower than what they would have been had the company consistently adopted the relevant counterfactual.

**Economic impact and earnings at risk**

Carbon price data are from the World Bank and are available at https://carbonpricingdashboard.worldbank.org/. We include any country with an active Emission Trading Scheme (ETS) during the observation period. For countries with both national and regional ETSs we use the price of carbon from the national market. For countries with only regional (or state) ETSs, such as the US, for example, where only a few states have active carbon pricing mechanisms, we use the average carbon price as national carbon price. For countries in the European union we use carbon prices from the EU ETS. The economic impact of the methane conversion gap for company $c$ in year $t$ is calculated as the product of the conversion gap and the price of carbon:

$$\mathcal{C}^{H}_{CH_4,c}(t) = \Delta^{H}_{CH_4,c}(t) \times \text{Carbon Price}(t) \qquad (4)$$

From the above, we can compute useful metrics, such as the Earnings at Risk (EAR):

$$\text{EAR}_c(t) = -\frac{\mathcal{C}^{H}_{CH_4,c}(t)}{\text{EBITDA}_c(t)}, \qquad (5)$$

where EBITDA are Earnings Before Interests, Taxes, Depreciation and Amortisation as reported in COMPUSTAT. EBITDA and carbon prices are measured in US\$ and US\$/tCO$_2$e, respectively. The transition risk analysis covers 1098 out of the 1718 companies in countries with available carbon prices because only for these companies we were able to match accounting data.

## Data availability
CDP data are covered by a license and therefore we cannot share them. Access can be obtained directly from the data providers for a fee (https://www.cdp.net/en).

## Code availability
The Python code to reproduce our results is available on Harvard Dataverse at https://doi.org/10.7910/DVN/ISMPPZ.

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

## Acknowledgements

The authors gratefully acknowledge the support of the Singapore Green Finance Centre (SGFC) and Giuseppe Brandi and Robin Lamboll for valuable feedback on a preliminary draft of the manuscript.

## Author contributions

SC and EB designed the study. SC collected the data and performed the analysis. SC and EB wrote the manuscript.

## Competing interests

The authors declare no competing interests.
