## [Transparent Peer Review file · Nature Communications]

Lack of harmonisation of greenhouse gases reporting standards and the methane emissions gap

Corresponding Author: Dr Simone Cenci

This manuscript has been previously reviewed at another journal. This document only contains information relating to versions considered at Nature Communications. Mentions of the other journal have been redacted.

Version 0:

Reviewer comments:

Reviewer #1

(Remarks to the Author)

I would like to thank the authors for taking the time to thoroughly revise the paper and thoughtfully consider my suggestions. I believe that both the inclusion of carbon pricing across more geographical contexts and the expanded discussion have substantially improved the quality of the paper. In this revised version, I have only a few remaining comments.

1. The authors mention in the results section that companies in the Energy, Utilities, and Materials sectors dominate the conversion gap and that economic losses are largely driven by firms in these sectors. I am concerned about methane emissions, which are significantly sourced from agricultural activities (see <https://ourworldindata.org/grapher/methane-emissions-by-sector>). In both the main text and Table ST4, the company sample appears to exclude agriculture and related sectors, which may represent an omission. Could the authors clarify whether this is indeed the case, and if so, discuss the potential implications of not including these sectors in the analysis?

2. Lines 168-171: Please specify that “170 MtCO₂e” and “3300 MtCO₂e” represent cumulative emissions, as well as clarify this in other relevant sections. Without this distinction, readers may misinterpret these figures, thereby exaggerating the conversion gap.

(Remarks on code availability)

Reviewer #2

(Remarks to the Author)

(Remarks on code availability)

Reviewer #3

(Remarks to the Author)

I reviewed this paper during its initial submission to **Redacted** and appreciate the significant efforts the authors have made to improve the manuscript. For this revised version, I recommend the following changes prior to publication:

Major Comments

1. The central message of the manuscript emphasizes the importance of globally harmonized CO₂-equivalence standards

for consistent tracking of corporate greenhouse gas emissions—a valuable policy suggestion, which makes it more fitting as a ‘Perspective’ article rather than a research-focused piece. Ultimately, this decision lies with the editor after considering feedback from other reviewers.

2. The manuscript includes excessive repetition, leading to redundancy. For instance, the abstract is overly long, and the introduction spans seven paragraphs, many of which reiterate similar points. This lack of conciseness makes it challenging to follow the argument’s progression. I strongly suggest streamlining these sections to ensure clarity and maintain reader engagement.

3. The cumulative reporting in Figure 2 is misleading, as the values grow proportionally with the integration period, exaggerating the effects. A similar issue appears in Figure 4. I recommend presenting data on an annual basis unless the authors can provide a strong rationale for the cumulative approach.

Minor comments.

Line 11-13. Please also report methane emissions in Tg a-1. If I use a GWP of 28 to convert from CO2e to methane emissions, the number will be 6 Tg a-1.

Lines 118-124. Well, this belongs to the introduction.

Line 125-137. The samples presented in this work represent 2.5% of global methane emissions. Is this small fraction truly significant for assessing the global methane budget? Clarifying its importance or providing additional context would strengthen the argument.

Line 152. How much is the recent GWP100? Needs to be specific in the main text to facilitate the reading?

Line 168-171. Similarly, please specify the GWP values applied in your calculations. Explicitly stating these values will ensure transparency and allow readers to accurately interpret the results.

(Remarks on code availability)

The authors have provided the code, but I have not executed it to verify its functionality.

Reviewer #4

(Remarks to the Author)

The manuscript analyses the impact of the optionality in the choice of the global warming potential for data voluntarily reported by companies to the CDP. It reads well and the method is simple but sound. I only see a minor flaw which is that the authors tend to mix up two issues, the impact of “optionality” (when they compare actual data with their “GWP100 counterfactual”, representing the dominant norm, once harmonized to the latest IPCC values) and the impact of changing the dominant norm of the GWP100 to a GWP20 (all figures related with the “GWP20 counterfactual”). Provided that the distinction between these two different issues is clarified in the abstract and in the manuscript (see examples below), I recommend that the manuscript be accepted for publication.

L11: “Using a 20-year GWP, as recently codified in certain jurisdictions and initiatives, makes the gap grow to 3300MtCO₂e. The gap only covers direct (Scope 1) emissions and hence understates the potential extent of under-reporting across value chains, particularly in the Energy sector.”

Because this follows the state purpose that “we illustrate the implications of reporting optionality for monitoring corporate emissions” (18), one implicitly understands that the quoted sentences still serve this purpose, which is not the case. As argued above and below, the analysis using the GWP20 counterfactual simulates a change in the dominant norm rather than the suppression of “optionality”. It should be very clear in the abstract, for example by changing the first quoted sentence to “Changing the dominant norm of the 100-year GWP to a 20-year GWP, as recently codified in certain jurisdictions and initiatives, would increase reported emissions by 3300MtCO₂e.”

L257: “we find that for the Energy, Material and Utilities sectors companies face transition risks that can be as much as 25% higher than currently reported moving from 13% to 38% of total earnings under the GWP20 counterfactual.”

Highlighting only the results under the GWP20 counterfactual is misleading. As the authors themselves acknowledge, GWP100 counterfactual is the most appropriate as GWP100 is the dominant norm. I would recommend mentioning the results under the GWP100 counterfactual before mentioning the results under the GWP20 counterfactual, and reminding the reader that results under the GWP20 counterfactual simulate a change in the dominant norm rather than the suppression of “optionality” or the fixing of “Loose guidelines and regulations”.

L334 “Optionality in the choice of emission metrics has important implications for investors assessing their portfolios’ exposure to transition risks, as proxied, for example, by carbon prices. Indeed, we have found that if companies were to follow harmonised standards, the cost of offsetting their methane emissions would represent a sizeable portion of their earnings. The effect is particularly evident in the Energy and energy-intensive sectors, in which, under the GWP20 counterfactual, earnings at risk can be as high as four times those implied by reported methane emissions.”

Same issue here “The effect” mentioned in the last sentence is not the effect of suppressing “optionality” or “following harmonised standards”, as implied by the first two sentences, but it is mostly the effect of changing the norm from GWP100 to GWP20.

(Remarks on code availability)

The images or other third party material in this Peer Review File are included in the article’s Creative Commons license, unless indicated otherwise in a credit line to the material. If material is not included in the article’s Creative Commons license and your intended use is not permitted by statutory regulation or exceeds the permitted use, you will need to obtain permission directly from the copyright holder.

Response to Reviewers: Lack of harmonisation of greenhouse gases reporting standards and the methane emissions gap

Reviewer 1

1. I would like to thank the authors for taking the time to thoroughly revise the paper and thoughtfully consider my suggestions. I believe that both the inclusion of carbon pricing across more geographical contexts and the expanded discussion have substantially improved the quality of the paper. In this revised version, I have only a few remaining comments.

Answer. Thank you for reviewing our manuscript again and providing valuable suggestions. We have addressed all your remaining concerns, as detailed in our responses below.

2. The authors mention in the results section that companies in the Energy, Utilities, and Materials sectors dominate the conversion gap and that economic losses are largely driven by firms in these sectors. I am concerned about methane emissions, which are significantly sourced from agricultural activities (see <https://ourworld-indata.org/grapher/methane-emissions-by-sector>). In both the main text and Table ST4, the company sample appears to exclude agriculture and related sectors, which may represent an omission. Could the authors clarify whether this is indeed the case, and if so, discuss the potential implications of not including these sectors in the analysis?

Answer. Our sample includes companies in the agricultural sector, which is part of the Consumer Staples sector (see table ST4 and the new comment in Figure S5). The Consumer Staples sector as a whole represents 5% of the total sample, but the median emissions of companies in this sector are ~ 3 times as large as the median methane emissions of the rest of the universe of companies. By isolating only companies involved in agricultural and farming products¹ the subsample shrinks to $\sim 1.5\%$ of the total sample, but the median emissions of this subsample are ~ 300 times larger than the median methane emissions of companies in other sectors. This confirms that in our sample agricultural firms are large methane emitters, but we only have a handful of them. Note that, as now discussed in line 227-228 of the new version of the manuscript, several agriculture firms are not included in the economic impact analysis because they are located in countries without an active ETS in the observation period. This further illustrates how conservative our estimates are and underscores the urgency to improve regulations around reporting of disaggregated GHGs.

¹We look at companies' primary activities as classified by CDP and retain those involved in 'Dairy & egg products', 'Cattle farming', 'Agricultural products wholesale', 'Other crop farming', 'Other oilseed farming', 'Fruit farming', 'Sugarcane farming', 'Grain & corn farming', 'Soybean farming', 'Palm oil farming', 'Cotton farming', 'Poultry & other animal farming', 'Rice farming', 'Animal processing', 'Other animal farming and processing', 'Palm oil processing', 'Poultry & hog farming'

-
3. Lines 168-171: Please specify that “170 MtCO_{2e}” and “3300 MtCO_{2e}” represent cumulative emissions, as well as clarify this in other relevant sections. Without this distinction, readers may misinterpret these figures, thereby exaggerating the conversion gap.

Answer. Agreed, thank you for the suggestion. We have now reinforced this important point in the abstract and throughout the manuscript; see for example lines 154, 156, and 266.

Reviewer 2

Answer. Thank you for taking the time to review our manuscript. The useful feedback received was instrumental in strengthening the clarity and potential impact of our work.

Reviewer 3

1. I reviewed this paper during its initial submission to **Redacted** and appreciate the significant efforts the authors have made to improve the manuscript. For this revised

version, I recommend the following changes prior to publication:

Answer. Thank you again for taking the time to review our work. We appreciate your comments and have addressed your concerns in the revised manuscript, as detailed further below.

2. The central message of the manuscript emphasizes the importance of globally harmonized CO₂-equivalence standards for consistent tracking of corporate greenhouse gas emissions—a valuable policy suggestion, which makes it more fitting as a ‘Perspective’ article rather than a research-focused piece. Ultimately, this decision lies with the editor after considering feedback from other reviewers.

Answer. We agree that the central message of the manuscript is to emphasize the importance of globally harmonised CO₂-equivalence standards for consistent tracking of corporate greenhouse gas emissions. However, we have also performed an extensive analysis of the largest dataset we could assemble to date and investigated the mechanics and dynamics of lack of harmonisation to substantiate our main message. We characterised the structure of the gap, its intertemporal dynamics and the implication for companies’ exposure to transition risks. Hence, we believe that our work classify as a research-focused piece but we clearly agree that ultimately the decision lies with the editor and the editorial team.

3. The manuscript includes excessive repetition, leading to redundancy. For instance, the abstract is overly long, and the introduction spans seven paragraphs, many of which reiterate similar points. This lack of conciseness makes it challenging to follow the argument’s progression. I strongly suggest streamlining these sections to ensure clarity and maintain reader engagement.

Answer. Thank you for the suggestions. We have re-arranged some statements, streamlined content and removed redundancies in the Introduction as well as in the Results section, as suggested in point 6 below. As for the Abstract, we have shortened it, while making sure it still contains information relevant to properly summarise our work.

4. The cumulative reporting in Figure 2 is misleading, as the values grow proportionally with the integration period, exaggerating the effects. A similar issue appears in Figure 4. I recommend presenting data on an annual basis unless the authors can provide a strong rationale for the cumulative approach.

Answer. The reason why we want to clearly illustrate the cumulative effect is that it represents how much it has been cumulatively lost in methane emission tracking in the past decade. However, we agree that a comprehensive picture of the evidence should include results presented on an annual basis to avoid any misunderstanding. To that effect, we have produced an improved Figure 2, which now includes: (a) cumulative effects (panels **a, e, c, g**), (b) distribution across the sample (firm-year) by sector (panels **b, f**), (c) yearly average gap across the population (panels **d, h**, left y-axis, gray line), (d) yearly average gap by sector (panels **d, h**, left y-axis) and (e) yearly total gap (panels **d, h**, right y-axis). Following your suggestion, Figure 2 now provides a considerably more comprehensive view of the gap, including cumulative and yearly values.

5. Line 11-13. Please also report methane emissions in Tg a⁻¹. If I use a GWP of 28 to convert from CO₂e to methane emissions, the number will be 6 Tg a⁻¹.

Answer. Thank you for the suggestions, we have now reported the values also in Tg a⁻¹ (see the new version of the Abstract). The values are an approximation because the reference GWP changed slightly during the observation period.

6. Lines 118-124. Well, this belongs to the introduction.

Answer. Point well taken. We have now moved this discussion to the Introduction, lines 71-81 (as already discussed in response to point 3. above). Thank you for the suggestion.

7. Line 125-137. The samples presented in this work represent 2.5% of global methane emissions. Is this small fraction truly significant for assessing the global methane budget? Clarifying its importance or providing additional context would strengthen the argument.

Answer. It is important to remark that in this work we are not attempting to assess the global methane budget. Instead, we are investigating the extent to which direct methane emissions may go under reported due to lack of harmonization in reporting standards (lines 82-87 and 257-261 of the new version of the manuscript). Clearly, these reporting activities will ultimately impact the estimation of a residual methane budget, but we agree, this can be done only to the extent that our estimates are extrapolated to a substantially larger sample. Unfortunately, data on methane emissions are rarely available and therefore such an exercise would require substantial changes in regulatory frameworks to impose strict reporting requirements on different GHG emissions.

8. Line 152. How much is the recent GWP100? Needs to be specific in the main text to facilitate the reading? Line 168-171. Similarly, please specify the GWP values applied in your calculations. Explicitly stating these values will ensure transparency and allow readers to accurately interpret the results.

Answer. Figure 1 (panel a) shows the values we used in this work, which are now also signposted in lines 142-152 of the revised manuscript. For completeness, the values and their sources are also reported in table ST2 in the Supplementary Information. Thank you for bringing to our attention this point requiring greater clarity.

Reviewer 4

1. The manuscript analyses the impact of the optionality in the choice of the global warming potential for data voluntarily reported by companies to the CDP. It reads well and the method is simple but sound. I only see a minor flaw which is that the authors tend to mix up two issues, the impact of “optionality” (when they compare actual data with their “GWP100 counterfactual”, representing the dominant norm, once harmonized to the latest IPCC values) and the impact of changing the dominant norm of the GWP100 to a GWP20 (all figures related with the “GWP20 counterfactual”). Provided that the distinction between these two different issues is clarified in the abstract and in the manuscript (see examples below), I recommend that the manuscript be accepted for publication.

Answer. Thank you for taking the time to review our manuscript, for the positive feedback, and for pointing out areas requiring more clarity. We have now taken onboard your suggestions as explained further below.

2. L11: “Using a 20-year GWP, as recently codified in certain jurisdictions and initiatives, makes the gap grow to 3300MtCO_{2e}. The gap only covers direct (Scope 1) emissions and hence understates the potential extent of under-reporting across value chains, particularly in the Energy sector.” Because this follows the state purpose that “we illustrate the implications of reporting optionality for monitoring corporate emissions” (18), one implicitly understands that the quoted sentences still serve this purpose, which is not the case. As argued above and below, the analysis using the GWP20 counterfactual simulates simulate a change in the dominant norm rather than the suppression of “optionality”. It should be very clear in the abstract, for example by changing the first quoted sentence to “Changing the dominant norm of the 100-year GWP to a 20-year GWP, as recently codified in certain jurisdictions and initiatives, would increase reported emissions by 3300MtCO_{2e}.”

Answer. Thank you for the suggestion. We have now made a few changes in the abstract to address this flaw. However, we would like to notice that changing from the GWP₁₀₀ to an *harmonised* GWP₂₀ implies both a change in norm and a “suppression of optionality”

3. L257: “we find that for the Energy, Material and Utilities sectors companies face transition risks that can be as much as 25% higher than currently reported moving from 13% to 38% of total earnings under the GWP20 counterfactual.” Highlighting only the results under the GWP20 counterfactual is misleading. As the authors themselves acknowledge, GWP100 counterfactual is the most appropriate as GWP100 is the dominant norm. I would recommend mentioning the results under the GWP100 counterfactual before mentioning the results under the GWP20 counterfactual, and reminding the reader that results under the GWP20 counterfactual simulate a change in the dominant norm rather than the suppression of “optionality” or the fixing of “Loose guidelines and regulations”.

Answer. Point well taken. We have now included a discussion of the results under the GWP₁₀₀ and stressed that the GWP₂₀ numbers refer to a change of norm, see line 248 of the new version of the manuscript.

4. L334 “Optionality in the choice of emission metrics has important implications for investors assessing their portfolios’ exposure to transition risks, as proxied, for example, by carbon prices. Indeed, we have found that if companies were to follow harmonised standards, the cost of offsetting their methane emissions would represent a sizeable portion of their earnings. The effect is particularly evident in the Energy and energy-intensive sectors, in which, under the GWP20 counterfactual, earnings at risk can be as high as four times those implied by reported methane emissions.” Same issue here “The effect” mentioned in the last sentence is not the effect of suppressing “optionality” or “following harmonised standards”, as implied by the first two sentences, but it is mostly the effect of changing the norm from GWP100 to GWP20.

Answer. Thank you for the suggestion. In line 326 of the new version of the manuscript, we have now clarified the meaning of "the effect", which refers to the effect of changing the norm to an harmonised standard (either GWP₁₀₀ - panel f - or GWP₂₀ - panel g). Overall, we would like to thank you for pointing out that the distinction between the impact of optionality and a change in relevant norm required clarification.